# The NDV-MLS as an Immunotherapeutic Strategy for Breast Cancer: Proof of Concept in Female Companion Dogs with Spontaneous Mammary Cancer

**DOI:** 10.3390/v16030372

**Published:** 2024-02-28

**Authors:** Diana Sánchez, Gabriela Cesarman-Maus, Laura Romero, Rogelio Sánchez-Verin, David Vail, Marina Guadarrama, Rosana Pelayo, Rosa Elena Sarmiento-Silva, Marcela Lizano

**Affiliations:** 1Unidad de Investigación Biomédica en Cáncer, Instituto Nacional de Cancerología, Mexico City 14080, Mexico; 2Departamento de Medicina Genómica y Toxicología Ambiental, Instituto de Investigaciones Biomédicas, Universidad Nacional Autónoma de México, Mexico City 04510, Mexico; 3NorthStar VETS, Veterinary Emergency Trauma & Specialty Centers, Robbinsville, NJ 08691, USA; 4Departamento de Hematología, Instituto Nacional de Cancerología, Mexico City 14080, Mexico; gcesarman@gmail.com; 5Departamento de Patología, Facultad de Medicina Veterinaria y Zootecnia, Universidad Nacional Autónoma de México, Mexico City 04510, Mexico; lromeror@unam.mx (L.R.); marinago@fmvz.unam.mx (M.G.); 6Departamento de Patología, Hospital Ángeles Pedregal, Mexico City 10700, Mexico; genryus_blade@hotmail.com; 7Department of Medical Sciences, School of Veterinary Medicine, University of Wisconsin, Madison, WI 53706, USA; david.vail@wisc.edu; 8Unidad de Educación e Investigación, Instituto Mexicano del Seguro Social, Mexico City 06720, Mexico; rosana.pelayo.c@gmail.com; 9Centro de Investigación Biomédica de Oriente, CIBIOR, Instituto Mexicano del Seguro Social, Puebla 06720, Mexico; 10Departamento de Microbiología e Inmunología, Facultad de Medicina Veterinaria y Zootecnia, Universidad Nacional Autónoma de México, Mexico City 04510, Mexico; rosass@unam.mx

**Keywords:** Newcastle disease virus, canine mammary tumors, oncolytic virotherapy, canine cancer, breast cancer immunotherapy

## Abstract

The absence of tumor-infiltrating lymphocytes negatively impacts the response to chemotherapy and prognosis in all subtypes of breast cancer. Therapies that stimulate a proinflammatory environment may help improve the response to standard treatments and also to immunotherapies such as checkpoint inhibitors. Newcastle disease virus (NDV) shows oncolytic activity, as well as immune modulating potential, in the treatment of breast cancer in vitro and in vivo; however, its potential to enhance tumor-infiltrating immune cells in breast cancer has yet to be evaluated. Since spontaneous canine mammary tumors represent a translational model of human breast cancer, we conducted this proof-of-concept study, which could provide a rationale for further investigating NDV-MLS as immunotherapy for mammary cancer. Six female companion dogs with spontaneous mammary cancer received a single intravenous and intratumoral injection of oncolytic NDV-MLS. Immune cell infiltrates were evaluated by histology and immunohistochemistry in the stromal, intratumoral, and peritumoral compartments on day 6 after viral administration. Increasing numbers of immune cells were documented post-viral treatment, mainly in the peritumoral compartment, where plasma cells and CD3+ and CD3-/CD79- lymphocytes predominated. Viral administration was well tolerated, with no significant adverse events. These findings support additional research on the use of NDV-MLS immunotherapy for mammary cancer.

## 1. Introduction

The success achieved with immunotherapy in human cancers such as melanoma, non-small-cell lung carcinoma, Hodgkin´s lymphoma, Merkel cell carcinoma, and renal cancer emphasizes the importance of the immune system in cancer eradication [1,2,3,4,5,6,7,8,9]. Immunotherapy has been a turning point of cancer treatment, significantly increasing clinical responses, quality of life, and survival [10]. Historically, breast cancer has been classified as immunologically silent, characterized by the presence of few effector tumor-infiltrating immune cells [11]. Several distinct populations of tumor-infiltrating lymphoid and myeloid cells affect the prognosis of breast cancer patients; thus, lesions that are devoid of tumor-infiltrating leukocytes present a major clinical challenge [12,13,14,15,16,17,18]. In this context, several approaches have been proposed to enhance the anticancer immune responses, including combinations of immune modulators with either conventional or novel strategies for breast tumors that have minimal or no immune infiltrates [19,20].

Oncolytic viruses have both cytotoxic and immune-stimulatory effects [19,20,21,22]. Among these viruses, the Newcastle disease virus (NDV) has demonstrated a selective cytotoxic effect on a wide variety of malignant mammalian cells, including those of canine mammary and human breast cancer [23,24,25]. NDV has been shown to induce tumor inflammatory responses and an antitumor effect in preclinical poorly immunogenic mouse models of solid tumors [26,27,28]; it has an acceptable safety profile in human patients, non-human primates, and as a vaccine vector in healthy dogs and cats [29,30,31,32]. Across clinical studies performed in humans, the intravenous administration of NDV has, in general, been well tolerated, with flu-like symptoms being the most common reported adverse events [33]. Since NDV can improve the tumor immune infiltrates, it may represent an additional strategy of immunotherapy for breast cancer [26,28]. In this study, we administered the oncolytic and avirulent NDV-MLS to female companion dogs with spontaneous mammary cancer as a proof of concept of the viral capacity to stimulate immune-cell infiltrates into the tumors. Of note, studies in companion dogs with spontaneous cancers represent a relevant parallel patient population which may overcome some of the limitations of studies performed in murine models [34,35,36]. Dogs are immunocompetent; their tumors occur in an autochthonous tumor microenvironment and better recapitulate the responses to treatments, as well as the complexity of human diseases, compared to models such as immunocompromised mice [37,38,39]. 

## 2. Materials and Methods

### 2.1. Study Design

Six female companion dogs with mammary cancer scheduled for conventional surgical treatment were enrolled. Six days before surgery, a baseline incisional tumor biopsy was obtained, followed by the administration of a single intravenous and intratumoral injection of NDV-MLS. Serum biochemistry panel (SBP), complete blood count (CBC), and coagulation test (PT/PTT) were performed at baseline and 6 days post-viral treatment. Vital sign parameters were recorded every 10 min for 1 h once viral administration was started, and dogs were then returned to the care of their owners. Dogs were monitored at home by their owners, and although NDV-MLS is a non-pathogenic strain, owners were instructed to follow safety measures to avoid any possible viral contact. Viral shedding through dog´s urine and saliva was evaluated at the recheck time on day 6. The development of antibodies against NDV and changes in plasma cytokines were also documented in samples at this time. On day 6, a tumor sample was taken, and tissues were analyzed through histopathology. The viral vehicle was injected intravenously and intratumorally to an additional female companion dog with mammary cancer who was scheduled for euthanasia, as a negative control. Vital parameters, blood and tissue sampling, and monitoring at home were performed similar to viral-treated dogs. None of the seven dogs had received non-steroidal anti-inflammatory drugs or cytotoxic agents for at least 1 month before viral administration or during the 6-day post-viral injection. Although in two dogs the surgical excision of the regional lymph node was indicated and performed as part of their treatment plan, these tissues were not included in the study, as we focused on the local tumor effect.

Our goal was to evaluate the potential of NDV to induce immune tumor infiltrates in canine patients with mammary cancer, without compromising their treatment regimen. Since Zamarin et al. (2014) previously found clear immune cell infiltrates in tumor-treated mice at day 8 post-initial viral treatment [26], and we previously noted tumor regression of a canine patient treated IV/IT after day 4 post-viral treatment (data not published yet), we considered day 6 as a reasonable time to evaluate tumor infiltrates post-viral treatment, which also represents a reasonable time between patient diagnosis, treatment plan discussion with the owner, and surgical procedure and/or additional treatment (e.g., Non-steroidal anti-inflammatory drugs). The day-6 post-viral treatment time point worked well for logistical reasons as well (coordination between days dedicated to viral treatment, surgery, and sample processing for pathology). 

All procedures were carried out under owner-approved informed consent and were carefully performed by veterinary medical professionals, ensuring dogs’ well-being. All experiments were approved by the Internal Committee for the Care and Use of Laboratory Animals (CICUAL) of the National Cancer Institute of Mexico (INCan), approved protocol (018/026/IBI) (CEI/1245/18) (2018/0705/CB1).

### 2.2. Newcastle Disease Virus

We used the NDV-MLS strain, a non-recombinant avirulent virus previously described as oncolytic [23]. Viral replication was carried out by inoculation of specific pathogen-free embryonated chicken eggs with 0.1 mL of NDV-MLS. After 3 days, allantoic fluid was aseptically removed from the eggs, and the presence of the virus was confirmed by micro-hemagglutination, using chicken red blood cells. The viral batch was tested for the presence of microbiological agents through cultures. NDV dose was expressed as the amount of mean infective doses in embryonated chicken eggs (EID50), as determined by infectivity assay and the Reed–Muench method [40]. Briefly, groups of six 9-day-old embryonated chicken eggs were inoculated with 0.1 mL of 10-fold dilutions of NDV-MLS. After 72 h of incubation at 37 °C, embryonated eggs were euthanized through cooling by refrigeration, and then the allantoic fluid was aseptically removed from eggs. The presence of the virus was confirmed by micro-hemagglutination, using chicken red blood cells. The embryonated eggs positive to infectivity were recorded, and the Reed–Muench formula was applied. The virus was aliquoted and stored at −70 °C until use. For administration, vials were thawed at a temperature of 4 °C and injected within 1 h of thawing.

### 2.3. Dogs

Female companion dogs were eligible if they had a malignant mammary gland tumor ≥ 3 cm longest diameter and had not been treated with cytotoxic agents, prednisone, or non-steroidal anti-inflammatory drugs over the last month. Other eligibility criteria included age older than 1 year; performance status defined as constitutional clinical signs grade ≤ 2 according to the Veterinary Cooperative Oncology Group-Common Terminology Criteria for Adverse Events (VCOG-CTCA v2) following investigational therapy in dogs and cats [41]; normal organ function, including a serum creatinine level < 1.5 times the upper limit of normal, bilirubin level < 1.5 times the upper limit of normal, and serum transaminase level < 2.5 times the upper limit of normal; and a calculated life expectancy of at least 1 month. Dogs with pregnant or nursing owners were not eligible, as well as dogs that had hypersensitivity to eggs, an autoimmune disease, any uncontrolled comorbidity, or an uncontrolled bacterial infection.

### 2.4. Viral Administration and Follow-Up

Prior to viral administration, basal samples were collected. Blood was obtained for determination of complete blood counts (CBCs), serum biochemistry profile (SBP), and coagulation test (PT/aPTT), as well as to evaluate the presence of NDV and NDV antibody titers and to measure plasma cytokines/chemokines. Urine and saliva were also collected to evaluate for the presence of infective viral particles. Tumors were marked with regularly spaced site points to facilitate biopsies and viral administration uniformly within the tumor. First, a tumor sample for histology was obtained from one of the marked points through an incisional biopsy (“slice of cake” technique) and with the dog under sedation. Immediately after all samples were collected, dogs received a single intravenous (1 × 106.88 EID50) and intratumoral (1 × 106.58 EID50) injection of NDV-MLS. The intravenous dose was infused over 40 min, along with 60 mL of 0.9% NaCl solution. The intratumoral injections were given with a 23-gauge needle at the remaining marked points. Needles were inserted into the tumor perpendicular to tumor surface, and each injection was given using continuous flow as the needle was withdrawn from the furthest point within the tumor. Dogs who had more than one mammary tumor received the intratumoral injection only in one of the tumors. Dogs were monitored every 10 min for one hour from the time viral infusion began. The parameters recorded included respiratory rate, heart rate, rectal temperature, and indirect blood oxygen saturation (sO_2_). Dogs were sent home if they did not show signs of illness during the viral administration and were clinically stable at least 1 h after the end of the viral infusion. Although NDV-MLS is an avirulent strain and NDV transmission between dogs and humans has not been documented, safety measures were followed. Personnel used personal protective equipment, including gloves, goggles, and lab coats. At home, owners kept the dog’s environment clean; feces and urine were inactivated with 10% sodium hypochlorite; and dogs did not have contact with underage or elderly individuals, pregnant woman, and individuals with immunodeficiency until 1 month after the virus was administered. Owners reported the dog’s condition daily directly to the veterinarian. Dogs were monitored for diarrhea, vomiting, coughing, sneezing, tumor inflammation, pain, or any other abnormal change in behavior. The assessment of adverse events was performed according to the Veterinary Cooperative Oncology Group—Common Terminology Criteria for Adverse Events VCOG-CTCAE (v2), following investigational therapy in dogs and cats [41]. Six days after the viral administration, dogs were again evaluated with a physical examination, including tumor measurement. Tumors were measured in their three dimensions (length, height, and width) with calipers, and the variations in the longest diameter in the plane of measurement were used to define tumor response per response evaluation criteria for solid tumors in dogs, RECIST (v1.0): a Veterinary Cooperative Oncology Group (VCOG) consensus document [42]. Tumor samples, blood, saliva, and urine were also collected at this time to repeat the studies performed at baseline. Biopsies at this point (day 6) were taken at an opposite/distant point from the day 0 tumor biopsy site and consisted of a “slice of cake” containing the needle track and surrounding tissue from the center to the periphery of the tumor.

### 2.5. Cytokine Measurement

Plasma cytokines (IL2, IL6, IL8, IL10, GM-CSF, MCP-1, RAGE, SCG, TNF-alpha, and VEGF) were measured before and 6 days post-viral administration in the plasma samples of dogs receiving the NDV-MLS, using a specific canine cytokine antibody array and according to the manufacturer’s instructions (Canine Cytokine Antibody Array A, Abcam ^®^, Cambridge, UK). Briefly, blocked glass slides spotted with the antibody arrays were added with samples and incubated at room temperature for 1.5 h. Then, samples were removed, and the slides were washed. The detection antibody cocktail was added to each well and incubated at room temperature for 2 h. After washing the slides, Cy3 equivalent dye-conjugated streptavidin was added and incubated at room temperature for 1 h. Finally, slides were washed and dried, and the signals were visualized through the use of a laser scanner with a Cy3 wavelength (GenePix 4100A microarray scanner, Molecular Devices, Sunnyvale, CA, USA). Samples were processed by quadruplicate, and the data extraction was performed with GenePix^®^ Pro 7.0 Image Analysis Software.

### 2.6. Virological Studies

The presence of active viral particles was ruled out through the evaluation of viral propagation in embryonated chicken eggs. Samples of urine, saliva, and plasma collected at baseline (day 0) and at day 6 post-viral administration were evaluated. A total volume of 0.1 mL of each sample was inoculated in 9-day embryonated chicken eggs. Three embryonated chicken eggs were inoculated per sample. After 72 h of incubation at 37 °C, the allantoic fluid was aseptically removed from eggs, and the presence of infective virus was tested by the detection of hemagglutination activity, using chicken red blood cells, and confirmed by specific reverse-transcriptase polymerase chain reaction (RT-PCR). The primers used for the RT-PCR were previously reported and comprised a conserved NDV nucleotide sequences of a fragment of 527 base pair (bp) of the nucleoprotein (NP) gene (FP: 5′-ACCAAACAGAG AATCCGTGAGTTACGATAA-3′; and RP: 5′-GGAGAGATCCTGCTATCATCGCAAATCT-3′) [23]. For reverse transcription (RT), 5 μL of extracted nucleic acid was added to the following reaction mix: 2 μL of 10 mM oligos, 0.5 μL of 10 mM dNTP mix, 2 μL of buffer, 0.5 μL of polymerase, and 1 μL of RNase inhibitor for a total reaction volume of 11 μL. As the positive control, 5 μL of NDV-MLS-extracted nucleic acid was used. For the negative control, only sterile water was added. The reaction was subsequently incubated in a thermo-cycler under the following conditions: 50 °C for 30 min and 95 °C for 15 min for RT; for PCR, 94 °C for 2 min, followed by 35 cycles at 94 °C for 30 s (denaturation), 55 °C for 30 s (annealing), and 72 °C for 1 min (extension). Samples were subsequently run on 2% agarose gel and photographed. In addition, the development of NDV-specific antibodies was also evaluated by hemagglutination inhibition assay (HIA), using pathogen-free chicken erythrocytes. Hemagglutination (HA) occurs when hemagglutinins on the virus envelope interact with receptors on the surface of erythrocytes, and inhibition of HA occurs in the presence of serotype-specific antibodies in the sample. Canine serum samples from baseline (day 0) and day 6 post-viral administration were inactivated by heating 30 min at 56 °C. Serial two-fold dilutions of serum samples were added with 4 hemagglutinating units (HAU) of NDV-MLS for 30 min at room temperature and then added with 1% (*v*/*v*) chicken erythrocytes. After gentle mixing, the chicken erythrocytes were allowed to settle for 40–60 min at room temperature until control of chicken erythrocytes settled to a distinct button. The serum end point of each assay was considered as the highest dilution of serum that prevents 100% of the hemagglutination of chicken red blood cells. Only those wells in which the chicken erythrocytes stream at the same rate as the control wells were considered for results. The HIA titer of individual serum samples was expressed as the inverse of the serum end point multiplied by the number of hemagglutinating viral units used. Positive serum of a previous NDV-vaccinated chicken, serum of a healthy dog, NDV-MLS, and pathogen-free chicken erythrocytes were used as controls.

### 2.7. Histopathology and Immunohistochemistry

Tumor samples obtained at baseline (day 0) and 6 days post-viral administration were processed for hematoxylin and eosin (H&E) staining, and for immunohistochemistry to identify infiltrating B- and/or T-lymphocytes. Samples were fixed in 10% formalin, paraffin-embedded, and then sectioned and stained with H&E. Immunohistochemistry for CD79a and CD3 was performed using immunoperoxidase according to the manufacturer’s recommendations (Mouse/Rabbit PolyDetector DAB HRP Brown Detection System, Bio SB, Santa Barbara, CA, USA). Briefly, tissue sections were cut at 4 μm, mounted on pre-cleaned and positively charged microscope slides, deparaffinized, and rehydrated. Antigen was retrieved by boiling slides in 0.01 M sodium citrate buffer, pH 6, and at 99–100 °C for 15 min (CD79a) or 20 min (CD3). Then, slides were removed from heat and allowed to stand in buffer for 30 min. Endogenous peroxidase was blocked with the PolyDetector peroxidase blocker, and then each slide was incubated with the corresponding primary mAb (1:200 CD79a and 1:50 for CD3) overnight at 4 °C in a wet chamber. Slides were washed with phosphate buffer (PBS pH 7.4) and were incubated with an anti-mouse secondary antibody (PolyDetector HRP Label) for 30 min. Positive staining was detected with 3-3′-diaminobenzidin PolyDetector Chromogen Buffer, and then the slides were counterstained with Harris’s hematoxylin (Merck, Darmstadt, Germany). Finally, slides were preserved with rapid mounting media (Entellan^®^ new, Merck, Darmstadt, Germany) and covered with a glass coverslip.

### 2.8. Histopathologic Evaluation

Two pathologists (R.S-V. and L.R.) performed two blinded evaluations. A first evaluation was performed independently, followed by a second side-by-side evaluation to reach a consensus. The two pathologists evaluated the slides and reported the immune cell infiltrates following recommendations provided by the international TILs working group [43]. Briefly, tumor samples were evaluated in the compartments: stromal, intratumoral, and peritumoral. Stromal infiltrates were defined as the cells located and dispersed in the stroma between the carcinoma cells and do not directly contact carcinoma cells. Intratumoral infiltrates were defined as immune cells in tumor nests having cell-to-cell contact with no intervening stroma and directly interacting with carcinoma cells. Importantly, both stromal and intratumoral compartments are localized in the region defined as tumor tissue. The peritumoral compartment was defined as tissues at the invasive edge, at least 100 µm outside the tumor border. Sections of 4–5 µm at a magnification of ×200–400 per patient were evaluated. The different compartments were evaluated in the same sample tissue. The sample technique performed allowed the pathologist to have all the compartments of the tumor in the same sample. All immune cells were scored (mononuclear and polymorphonuclear leukocytes). Immune cells were assessed as a continuous parameter. The percentage of immune cells in a particular compartment is a semiquantitative parameter; for example, “80% stromal lymphocytes” means that 80% of the stromal area shows a dense lymphocyte infiltrate. A full assessment of average cells in the tumor area by the pathologist was used, rather than focusing on hotspots. Immune cells in tumor zones with crush artifacts, necrosis or regressive hyalinization were excluded. The percentage of CD3- and/or CD79a-positive cells was reported using the percentage of the total lymphocytes observed in the H&E sections. Finally, the extents of necrosis, apoptosis, fibrosis, and cellular confluence were evaluated and scored in a continuous scale from 0 to 100%. For cellular confluence, we reported the changes noted vs. control sample; thus, the confluence at baseline was considered to be 100%.

### 2.9. Statistical Analysis

Data were analyzed using paired *t*-test in GraphPad Prism 10.0.2. Baseline (day 0) and post-treatment values (day 6 post-viral administration) of plasma cytokines levels and intratumoral cell infiltrates were compared, and differences were considered significant when *p*-values were <0.05.

## 3. Results

### 3.1. Simultaneous Intravenous and Intratumoral Injection of NDV-MLS to Female Companion Dogs with Spontaneous Mammary Cancer Was Well Tolerated and Did Not Result in Severe Adverse Events over the 6-Day Period Post-Administration

Signalment and tumor characteristics of dogs enrolled are summarized in Table 1. Baseline and 1-hour serial vital signs are presented in Figure 1. The parameters recorded included respiratory rate, heart rate, rectal temperature, and the indirect blood oxygen saturation (sO_2_). All six treated dogs showed transitory variations in cardiac and respiratory rates, but values were within the reference interval in five. In these five dogs, variations above baseline were attributed to stress due to medical manipulation. One dog identified as patient No. 6 showed a respiratory rate that doubled by minute 20 and remained elevated util minute 60. This dog showed a normal respiratory effort (eupneic) with no other signs and parameters normalized once the dog was at home. Regarding temperature and oxygen saturation, parameters were within normal limits in five of the six dogs. One dog (patient No. 1) had decreased sO_2_ (85%) at baseline, most likely attributed to lung metastasis, and the value declined by minute 50 (77–80%). This decrease in sO_2_ was not associated with other clinical signs, such as respiratory frequency alterations, respiratory distress, or abnormal color of mucosa. This dog continued to be monitored at home and did not develop any other clinical sign over the next 6 days. No other changes during the infusion period were documented, and none of the 6 dogs receiving the virus showed immediate reactions at the injected tumor.

After viral administration, dogs continued to be monitored at home for 6 days; the owners provided a daily update by phone directly to the veterinarian. On day 6, the dogs were evaluated with a physical examination, as well as a CBC, SBP, and coagulation test to monitor adverse events (AE). The serum chemistry for patient No. 4 could not be performed due to insufficient blood sample. Respiratory frequency at day 6 for dog 2 was not obtained, since she had heavy panting due to excitement.

At the physical exam on day 6, which included tumor measurement, all dogs (treated and control) had tumors in stable disease according to RECIST criteria in dogs (v1.0). The injected tumors were considered as the target lesions, and these tumors were measured in their three dimensions (length, height, and width) with calipers, and the variations in the longest diameter in the plane of measurement were used to define tumor response per RECIST criteria. Stable disease was defined as a less than 30% reduction or a greater than or equal to 20% increase in the longest diameter of the target lesion (data not shown). 

Regarding constitutional clinical signs, 2/6 dogs showed grade I AE (mild lethargy/fatigue/general performance), which lasted less than 3 days; weight loss grade 1 (<10%) was detected in 4/6 dogs; and fever grade I (39.5–40 °C) was documented in 3/6 dogs (2 at day 6 and 1 at day 2, post-virus administration). Anorexia grade I was reported in 4/6 dogs (coaxing or dietary change required to maintain appetite). Respiratory abnormalities were noted in 2/6 dogs: one dog (patient No. 5) had mild nasal discharge (grade I) for 3 days; and another dog (patient No. 1) showed a decrease in sO_2_ (from 84% to 77%, grade II), with no other associated clinical signs. The owners of patient No. 1 declined additional tests or procedures to further investigate the decrease in the sO_2_ due to the absence of other clinical signs and because the dog was doing well at home (Figure 2).

CBC parameters documented outside the reference range at day 6 post-viral administration were as follows. Three dogs (patients 1, 2, and 3) developed mild anemia (grade I), which was non-regenerative; two dogs (patients 3 and 4) developed mild neutrophilia; three dogs (patients 1, 2, and 4) developed monocytosis; one dog (patient 5) developed lymphocytosis; one dog (patient 1) showed eosinopenia; one dog (patient 3) had a decreased in the platelet count (grade I); and of four patients with decreased lymphocytes at baseline, the parameter remained decreased in two (patients 1 and 4) and normalized in two (patient 3 and 6). The control dog developed a mild regenerative anemia, her basal elevated neutrophil count normalized, and her basal eosinophilia improved (Figure 3).

The SBP parameters documented outside the reference range at day 6 post-viral administration for five of the six dogs (patients 1, 2, 3, 5, and 6) were as follows. One dog (patient 6) developed elevated urea, but her creatinine remained within normal limits, and one dog (patient 1) with elevated urea baseline remained with this value elevated; one dog (patient 5) developed elevated ALT (grade I); one dog (patient 2) developed elevated globulins, and one dog (patient 3) who had elevated globulins at baseline remained with this value elevated; one dog (patient 2) showed a mild increase in phosphorus, and another dog (patient 3) with mild elevated phosphorus at baseline remained with this value mildly elevated; a mild increase in sodium was noted in one dog (patient 3); and three dogs (patients 2, 3, and 6) showed a mild increase in chloride. The serum chemistry on day 6 was not performed on patient 4 due to an insufficient sample. The control dog only showed hyperproteinemia due to hyperglobulinemia at baseline and remained similar at day 6, with no other changes documented (Figure 4).

Coagulation tests (TP/aPTT) were performed in the six dogs receiving the virus. None of the dogs developed abnormalities of coagulation parameters (Table 2). In addition, virological studies were performed to evaluate the presence of infective viral particles in urine, saliva, and plasma samples through viral propagation in embryonated chicken eggs. All samples from the six dogs receiving the virus were negative for hemagglutinating activity of the allantoic fluid to chicken red blood cells and confirmed by RT-PCR (data not shown). The development of antibodies against NDV was evaluated via a hemagglutination-inhibition test in day-6 serum samples: 2/6 dogs had developed antibodies between 32 and 64 hemagglutination units (Table 3).

### 3.2. The Administration of NDV-MLS to Companion Dogs with Spontaneous Mammary Cancer Resulted in an Increase in Tumor-Infiltrating Immune Cells and Changes in Plasma Pro-Inflammatory Cytokines

Injected tumors varied in size from 3.0 cm to 10.7 cm in diameter. In order to better distribute the virus throughout the tumor, the total dose was injected in 3–5 points and avoided the site of the baseline biopsy (day 0). A tumor tissue sample was also obtained on day 6 from all seven dogs; this sample was taken from an injected tumor site distant to the site of the baseline biopsy. In two dogs (patients 3 and 5), additional samples of virally injected tumor sites were also obtained and analyzed. These two patients underwent complete tumor excision, which allowed us to have samples from more than one injected site. In particular, for patient 3, this was important for us to corroborate the possible effect between injected sites, as this dog had the biggest tumor (at least 50% bigger than the second in size). None of these samples was obtained from the site of the baseline biopsy. Tissues from the seven dogs were evaluated via histology and immunohistochemistry by two pathologists (R.S-V. and L.R.). The percentage of immune cell infiltrates in the stromal, intratumoral, and peritumoral compartments was reported, as well as the percentage of necrosis, apoptosis, fibrosis, and cellular confluence (Table 4 and Table 5). Overall, in most of the treated dogs, we noted an increased number of the immune cell infiltrates (Figure 5). This increase was more marked in the peritumoral compartment, although the changes (cells and percentages) were not homogeneous for all the dogs or compartments. Values were not statistically different in the control dog; however, they reached statistical significance in two of the treated dogs (Figure 6).

In regard to the plasma cytokines, plasma samples from the six dogs receiving the virus were evaluated in quadruplicate, using a specific canine cytokine antibody array. Baseline (day 0) and day 6 post-viral administration samples were compared, and significant changes were found in five dogs. Dog 1 did not have changes in cytokine levels; dog 2 had an increase in IL-6, IL-8, and MCP1; dog 3 had a decrease in IL-6 and SCF; dog 4 showed an increase in IL-6 and GM-CSF; dog 5 showed an increase in IL-6 and SCF, as well as a decrease in MCP1 and TNF-alfa; and dog 6 showed an increase in IL-6 and a decrease in SCF. Overall, IL-6 was the cytokine most consistently affected with an increase in four dogs and decrease in one dog. None of the dogs showed changes in the levels of IL-2, IL-10, RAGE, or VEGF (Figure 7).

## 4. Discussion

Breast cancer is one of the leading causes of death worldwide, and evidence supports the inclusion of immunotherapy as part of the therapeutic regimens [45,46,47]. Immunotherapy may be used for the promotion of tumor immune-cell infiltration, thus activating the tumor environment, rendering it more susceptible to additional treatments [48]. The oncolytic Newcastle disease virus has proved to have a cytotoxic effect in both breast cancer cell lines and mice models [25,49,50], as well as an immunostimulatory effect in tumors implanted in mice models [26]. No studies have been performed in immune-competent animal models such as dogs with spontaneous mammary cancer. It is worth mentioning that therapies such as these could be used in both human and veterinary medicine, since the immune response, disease behavior, and treatment responses are comparable [37,38]. 

In this study, we focused on the evaluation of the local response to a single combined (IT/IV) treatment of NDV-MLS. One of the challenges of oncolytic virotherapy has been the spread and adequate penetration of the virus into tumor tissue. To address this concern, multiple routes of administration have been evaluated. Responses to local (IT) or systemic treatment (IV) have been documented; however, the results are variable depending on the oncolytic virus and the type of cancer. In previous human clinical trials, a virulent oncolytic NDV promoted objective clinical responses after IV treatment [51,52,53,54], which highlights the possibility of this route to effectively deliver NDV to primary and metastatic lesions. On the other hand, NDV has been shown to promote immune infiltrates in solid tumors implanted in mice when treated IT [26], and more recently, the IT treatment of breast tumors in murine models with NDV was shown to inhibit tumor growth [50]. In our experience, the IV administration of NDV-MLS to a canine cancer patient was ineffective to deliver the virus to cancer tissue at 24 h post-injection; however, longer interval times or additional effects, such as the stimulation of circulating immune cells, were not assessed [23]. Moreover, another canine cancer patient treated with a single IT/IV dose of NDV-MLS achieved a favorable local immune response and clinical outcome (case report in progress). To maximize the possible effect of the NDV-MLS in inducing tumor immune infiltrates, we decided to approach this proof of concept by treating canine patients with a single simultaneous IT/IV treatment. We found that a single intravenous (1 × 106.88 EID50) and intratumoral (1 × 106.58 EID50) injection had an acceptable safety profile in companion dogs with spontaneous mammary cancer; it promoted diverse changes systemic cytokine levels, and the virus favored an increase in different immune cells infiltrating the tumors. In future studies, it may be necessary to elucidate if both routes are necessary or to determine if the effect relies only or primarily on one of them. 

We documented an acceptable safety profile to NDV-MLS treatment, which is supported by the lack of moderate or severe (local or systemic) adverse events (AEs), as well as for the absence of viral shedding into the environment. In this study, dogs receiving viral treatment showed almost only mild/grade I adverse events (AEs), which included infusion reactions, constitutional, respiratory, gastrointestinal, hematological, and biochemical changes, as well as effects on the tumor site. Infusion reactions included transitory changes in the respiratory and cardiac rates, as well as transitory oxygen desaturation (this was the only grade II AE reported). The alteration in the respiratory and cardiac rates were associated with medical manipulation; however, discomfort from the IV infusion cannot be ruled out. Human patients have reported transitory back pain and chest pressure during IV infusion of avirulent NDV; these symptoms were less common in humans when the doses and rate of administration were reduced [55]. Oxygen desaturation has been previously documented only in human patients with lung/pleural tumor involvement [51], as was observed in one dog who had lung metastasis. Other AEs included nasal discharge, lethargy/fatigue, fever, anorexia, and weight loss. Preclinical and clinical trials using NDV in human patients have documented flu-like symptoms, with fatigue and fever being the most common. These AEs are expected and known to be related to the release of pro-inflammatory cytokines [51,53]. Moderate–severe symptoms are treated in humans with combined antipyretics, and improved tolerability was noted with repeated viral doses and with the use of desensitization doses [53]. The weight loss and anorexia (all grade I) could be attributed to the discomfort caused by treatment and/or disease progression. Although previous human trials have documented grade 1–3 gastrointestinal AEs, weight loss has not been evaluated in humans [51,52,54,56]. Injected tumors of all dogs (treated and control) remained in stable disease. Although tumor size change is used as an efficacy metric for cancer treatment, in immunotherapy, changes in the tumor size can be temporally varied and may undergo pseudoprogression. In this study, we evaluated tumor size according to the RECIST criteria, with the main interest of documenting the behavior of the canine mammary tumors to this particular immunotherapy. Future studies should assess response based on newer immune-RECIST criteria that include a conformation time point. Changes noted on CBC at day 6 post-virus administration were mild and included mild non-regenerative anemia in three dogs and mild thrombocytopenia in one. None of these changes were clinically significant, as has been reported in humans, in whom these hematological effects are not dose dependent and are often reduced or absent during subsequent cycles [52,54]. On the other hand, serum biochemistry performed at day 6 post-viral administration showed increases in several parameters, such as urea, ALT, total proteins, globulins, phosphorus, sodium, and chloride; however, changes were mild, and none was clinically relevant. In humans, elevation in liver transaminase has been commonly reported and was transitory even in those with grade 3 elevation and despite receiving repeated doses [52,54]. The only AE noted on the tumor site was mild bruising at the periphery of the injected tumor in 4/6 dogs (Figure 8). All the dogs had normal coagulation tests and did not develop any other associated clinical signs. Avirulent NDV has not been previously injected in mammary tumors, and studies on intratumoral injection of other solid tumors in mice model have not reported this reaction [26,57,58]. It may be possible that, as happens in birds naturally infected with some NDV strains [59,60], the virus could damage blood vessel endothelium within tumor tissue. The latter needs additional investigation. Importantly, no infective viral particles were detected in the urine, saliva, or plasma at the end of the study time, consistent with avirulent strains which have limited replication capabilities; thus, viral shedding is not expected [61,62]. As was foreseeable, none of the dogs was seropositive at the start of the study, and two had developed NDV-specific antibodies by the 6th day post-virus administration. In human trials, the development of antibodies had no apparent relationship with the dose level [54], and signs of efficacy were observed in patients even after the formation of neutralizing antibody titers [51]. 

NDV-MLS treatment induced changes in the peripheral leukocytes and in the plasma pro-inflammatory cytokines and an increase in the tumor-infiltrating immune cells, which were significantly different among the dogs. By CBC, an increase in neutrophils, lymphocytes, and/or monocytes was observed in 5/6 dogs, with monocytes having the greatest increase (three dogs, >3× compared to baseline); however, these changes did not reach a statistical difference. In birds, an increase in the percentage of monocytes since day 1 post-NDV vaccination has been reported, which has been associated with monocyte activation by IFN-gamma released by Th cells upon infection [63]. In humans, systemic immune activation dependent on IFN-gamma has been shown to improve the antitumor activity of monocytes/macrophages [64,65] and to increase tumoricidal capacity of monocytes after the stimulation by NDV in vitro [66]. The clinical relevance of the increase in circulating monocytes post-NDV MLS treatment also requires further investigation. Regarding plasma cytokines, we found significant changes in 1-3 cytokines per dog in five of the six dogs. One patient (dog 1) did not show significant changes in any of the plasma cytokine levels. IL-6 was the cytokine most consistently affected, increasing in four dogs. Other changes involved the increase in IL-8, MCP1, SCF, and GM-CSF; and a decrease in MCP1, TNF-alfa, and SCF. None of the dogs showed changes in the levels of IL-2, IL-10, RAGE, or VEGF. IL-6 is a multifunctional cytokine that plays both pro-inflammatory and anti-inflammatory roles and has three signaling pathway routes [67]. IL-8 is a pro-inflammatory chemokine that recruits leukocytes to sites of infection or tissue injury [68]. MCP1 (monocyte chemoattractant protein-1) is an important chemokine which plays a crucial role in a number of pathological conditions and activates the signaling pathway which regulates the migration of cells [69]. SCF (stem cell factor) exerts biological functions by binding to and activating the receptor tyrosine kinase c-Kit, thus mediating cell survival, migration, and proliferation [70]. GM-CSF (granulocyte-macrophage colony-stimulating factor) is a cytokine that drives the generation of myeloid cells subsets, including neutrophils, monocytes, macrophages, and dendritic cells, in response to stress, infections, and cancer [71]. TNF (tumor necrosis factor)-alfa plays important roles in various biological processes, such as immunomodulation, fever, inflammatory response, response to tumors, and inhibition of virus replication [72]. All of these cytokines may play different roles depending on the cell type, as well as the normal or pathological condition; thus, these changes deserve further characterization.

Finally, we noted an increase in the tumor immune cell infiltrates in the six treated dogs, which were significantly different in two (dogs 3 and 5). Although we cannot completely rule out an influence of the basal biopsy in the tumor inflammatory process, this seems less likely or to have minimally contributed to the responses noted due to the site, size, type, and characteristics of the tissue samples. Overall, the changes were not homogenous but included an increase in lymphocytes (peritumoral, intratumoral, and stromal), macrophages (stromal, intratumoral, and peritumoral), neutrophils (stromal and intratumoral), and plasma cells (intratumoral and peritumoral). Of note, the control dog did not show any increase in immune cell infiltrates post-administration of the viral vehicle, and one patient (dog 6) who had no tumor immune cell infiltrate showed infiltration of intratumoral neutrophils, peritumoral lymphocytes, and peritumoral plasma cells. Breast/mammary cancer immunogenicity is highly heterogenous, with different subtypes showing different degrees of tumor immune infiltration [73,74]. In humans, the majority of the subtypes is considered to be low immunogenic [75], and similar to other tumor types, the tumor microenvironment (TME) plays a crucial role in treatment response. Tumors with higher immune cell infiltration and strong immune response tend to have better clinical outcomes [73]. Of note, the different tumor-infiltrating immune cells may play different roles, and for some of them, this is not well understood, such as for the neutrophils [76]. In the case of the plasma cells infiltrating breast cancer tumors (tumor-associated plasma cells, TAPCs), their relevance has been recently demonstrated. There is a positive correlation between TAPCs and outcome in triple-negative and hormone receptor-negative breast cancer patients; and a higher plasma cell infiltration in biopsy specimens before neoadjuvant chemotherapy was associated with pathological complete response [77,78,79]. In this study, an increasing number of plasma cells was documented in four of the six treated dogs, although their correlation with outcome needs additional investigation. Regarding tumor-associated macrophages (TAMs), these are usually recognized as drivers of tumor progression due to their immunosuppressive properties within tumors [80]. In breast cancer, a high density of TAMs is associated with a poor survival rate [81], and TAMs have been associated with more aggressive types of mammary cancer in dogs [82]. Activating or reprograming TAMs functions to pro-inflammatory phenotype to destroy tumor cells is one of the strategies to overcome their negative impact in cancer. In mouse models of cancer, TAM activation using Toll-like receptor (TLR) activators and CD40 agonist have been explored [83]. NDV has also been shown to activate macrophages to perform antitumor activities in vitro and in vivo in mice models of cancer, including mammary tumors [84,85]. Here, we documented an increasing number of TAMs in three of the six dogs, and although the difference did not reach statistical significance, this change may suggest a role of NDV-MLS activating dogs´ macrophages. This needs further evaluation, including characterization of macrophages’ phenotype. Finally, the presence of tumor-infiltrating lymphocytes (TILs) has been associated with a better prognosis in different subtypes of breast cancer [79,86], and emerging data suggest that TILs are associated with the response to both cytotoxic and immune therapies [87]. Therefore, the enhancement of the number of TILs and their activity against tumors is of great interest. In a poorly immunogenic mice model of melanoma, nonpathogenic NDV enhanced tumor infiltration with tumor-specific lymphocytes and had an antitumor effect in distant (non-virally injected) tumors without distant virus spread [26]; and infection of melanoma cells with avirulent NDV completely restored the proliferative response of T cells and inhibited the induction of anergy in vitro [88]. In this study, we found an increasing number of lymphocytes in the tumors of all six treated dogs; however, this change did not reach statistical difference in all. There was a predominant increase in the CD3+ and CD3−/CD79a− lymphocytes, suggesting an increase in T-cells and NK cells, respectively. NDV has previously been shown to activate NK cells in vitro, enhancing their ability to secrete effector lymphokines [89]. This possible effect and clinical implications in these patients need to be studied.

In our study, the two dogs with significant changes in the tumor immune cell infiltrates had tumors histologically classified as tubular carcinoma. As in humans, canine mammary tumors comprise a heterogeneous group of diseases where carcinomas are the most common histologic subtype. In regard to tubular carcinomas, this histology is rare in humans and nearly always positive for estrogen and progesterone receptors, as well as negative for HER2. These three markers are currently the most relevant clinical biomarkers widely used in stratifying human breast cancer case management. In dogs, immunohistochemical protocols have not been standardized for the evaluation of the same human molecular profiles; thus, although the same markers have been evaluated, the results have been variable and contradictory [90]. 

Since previous studies have demonstrated the cytotoxic effect of avirulent NDV on breast cancer cell lines [49,91,92,93], in a xenograft mouse mammary tumor model [50], as well as the previously discussed immunomodulatory properties of the virus, we evaluated on histology the necrosis, apoptosis, fibrosis, and cellular confluence as an indicator of these direct and/or indirect effects. Although we documented a decrease in the neoplastic cellular confluence in five of the treated dogs (dogs 1–5), this change did not reach statistical difference. It is possible that these changes may be more marked after additional doses or time, which needs additional investigation.

Our results support further research on NDV-MLS as an immunotherapeutic approach in breast cancer. The changes documented suggest a possible enhancement of the intratumoral immune response, which could be useful in combination with other immunotherapeutic protocols. The immune cell description we provided was limited, primarily because the phenotypic characteristics of immune cell populations in dogs are not completely established, in addition to the limited commercial reagents available. Subsequent studies should consider additional methods that help to elucidate the anti/pro-inflammatory nature of the immune cells and their possible anticancer effect. It will be important to study additional effects of viral treatment, such as the effect on type I IFN signaling pathways; the effect on non-injected distant tumors; alterations in the expression of surface molecules, such as MHC-II, adhesion molecules, and PD-L1/PD-1 signaling, as well as studying specific responses according to the molecular profiles of the tumor such that the translatability of the results will be more accurate and including additional controls and a larger sample size to completely rule out any possible effect of the biopsy in the tumor immune response.

NDV-MLS, as a lentogenic strain, is avirulent/low pathogenic to its permissive host (birds); thus, it is safe for the environment in the case of shedding. In addition, there is a lack of recombinant gene exchange and a lack of interaction with host cell DNA. Different strains of NDV have shown to be well tolerated in animal models and cancer patients, and as a non-genetically engineered virus, the cost of production is lower than other types of therapies, making it a financially accessible treatment.

## 5. Conclusions

A single simultaneous intravenous and intratumoral injection of the avirulent NDV-MLS was well tolerated in female companion dogs with mammary cancer, and the treatment caused an increase in tumor immune infiltrates. The magnitude of the different tumor-infiltrating immune cells post-viral treatment was not similar among dogs; this outcome may reflect tumor heterogeneity, as well as a pleiotropic effect of the virus. Our results support the establishment of additional research on the use of the NDV-MLS as a strategy to promote mammary tumor immunogenicity and/or modulate the TME. Studies in spontaneous mammary tumors in dogs are encouraged since they better recapitulate response to human treatment than other investigational models.

## Figures and Tables

**Figure 1 viruses-16-00372-f001:**
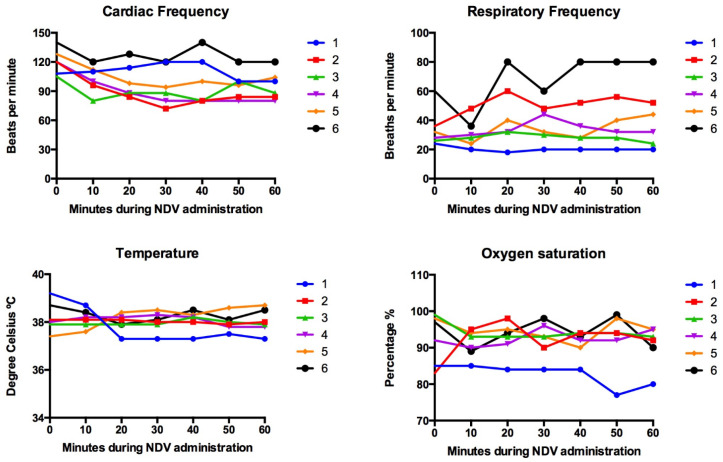
Simultaneous intravenous and intratumoral injection of NDV-MLS to companion dogs with mammary cancer did not cause clinically relevant alteration in their vital signs. Parameters were recorded during the hour following viral infusion. Each color line represents a dog. Dogs were identified as 1–6.

**Figure 2 viruses-16-00372-f002:**
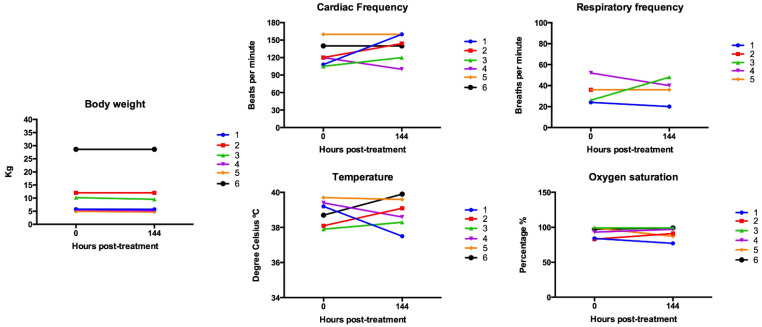
Physical parameters at baseline and day 6 after MDV-MLS administration. Respiratory frequency at day 6 for dog 2 was not obtained.

**Figure 3 viruses-16-00372-f003:**
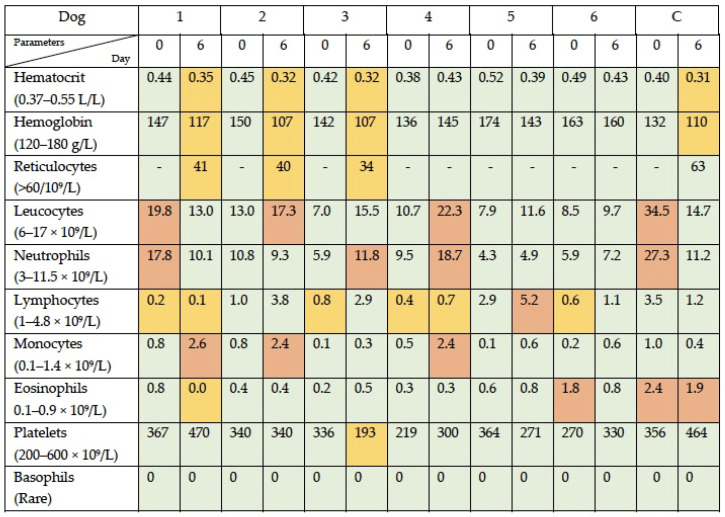
CBC data following simultaneous intravenous and intratumoral injection of NDV-MLS. Dogs were identified from 1 to 6. Control dog who only received the viral vehicle is included (C): 0, baseline; 6, day 6 post-viral administration. Green squares indicate the parameters within normal limits, yellow squares indicate the parameters below the reference range, and orange squares indicate the parameters above the reference range.

**Figure 4 viruses-16-00372-f004:**
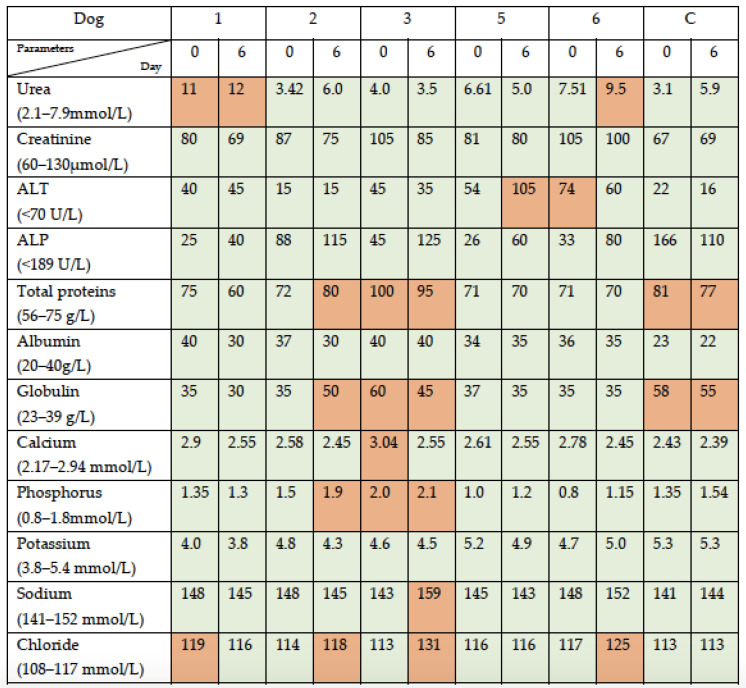
Serum biochemistry profiles (SBPs) following simultaneous intravenous and intratumoral injection of NDV-MLS. Dogs were identified from 1 to 6; results for dog 4 were not available. Control dog who only received the viral vehicle is included (C): 0, baseline; 6, day 6 post-viral administration. Green squares indicate the parameters within normal limits; orange squares indicate the parameters above the reference range.

**Figure 5 viruses-16-00372-f005:**
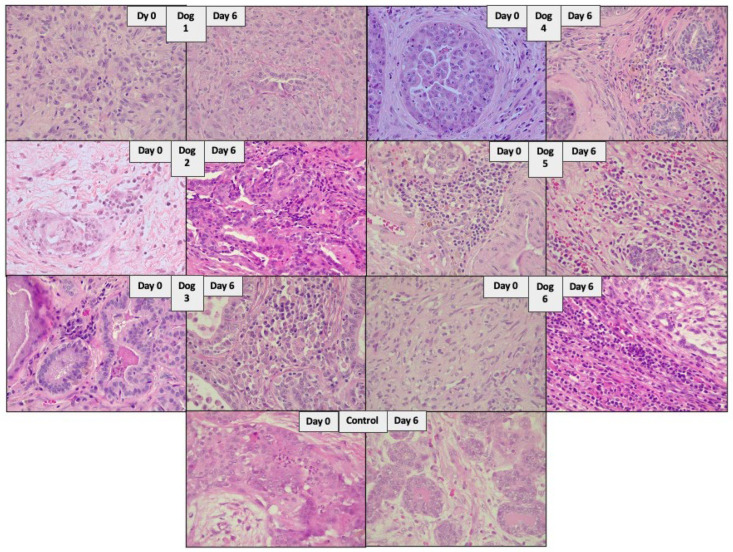
Representative microscopic images of tumor tissues of female companion dogs with spontaneous mammary cancer receiving the NDV-MLS (dogs 1–6). Images of the control dog receiving the viral vehicle are included (control). Images show H&E slides 40× of the tumor samples on day 0 and day 6 after viral administration.

**Figure 6 viruses-16-00372-f006:**
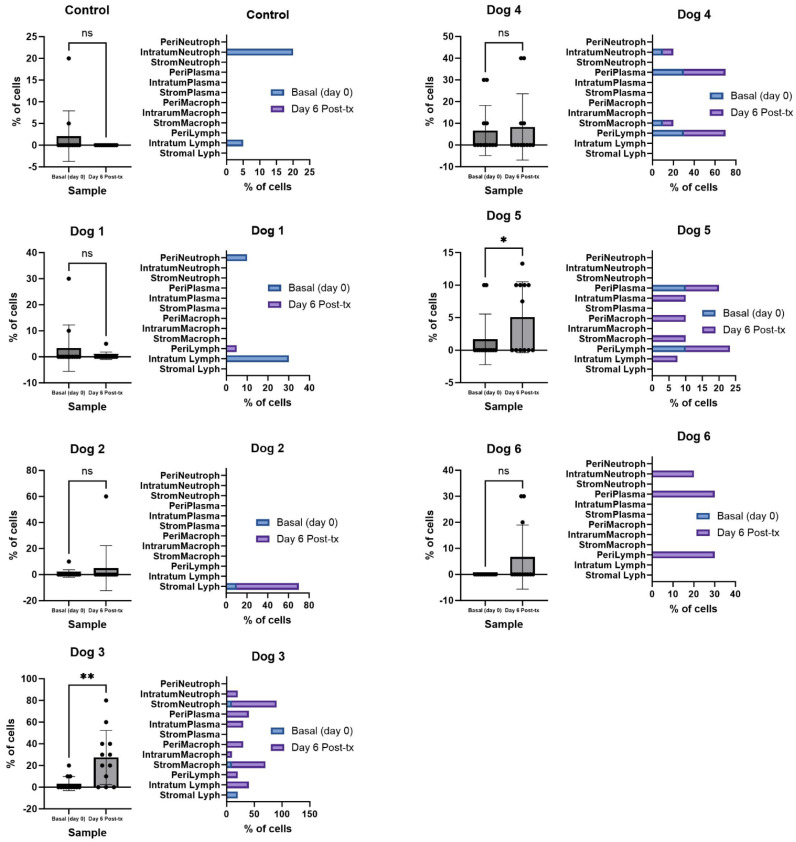
Variations in the tumor cell infiltrates after NDV-MLS administration. For each dog, the graphic on the left shows the differences between the tumor immune cells at baseline (day 0) vs. post-viral treatment (Day 6); and the graphic on the right shows the variation in the different cells and compartment. Peri, peritumoral; Intratum, intratumoral; Strom, stromal; Neutroph, neutrophils; Plasma, plasma cells; Macroph, macrophages; Lymph, lymphocytes; ns, non-significant; */** = *p* < 0.05. The tumor immune cell infiltrates at day 6 post-viral administration were statistically different in dogs 3 and 5, compared to their respective baseline (day 0) values.

**Figure 7 viruses-16-00372-f007:**
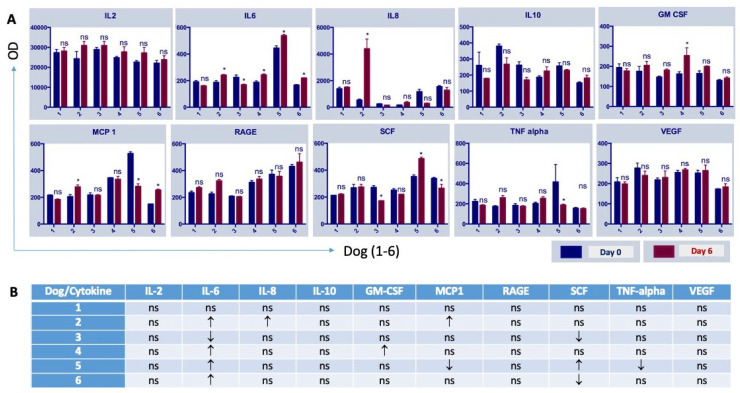
Variations in plasma cytokines after NDV-MLS administration. (**A**) Differences between baseline (day 0) and post-viral treatment (day 6) samples for individual plasma cytokines per dog are shown (* *p* < 0.05; ns, non-significant; OD, optical density). (**B**) Summary of the plasma cytokines variations (ns, non-significative changes; black arrows indicate the change between day 0 and day 6; arrow up means the value increased; arrow down means the value decreased).

**Figure 8 viruses-16-00372-f008:**
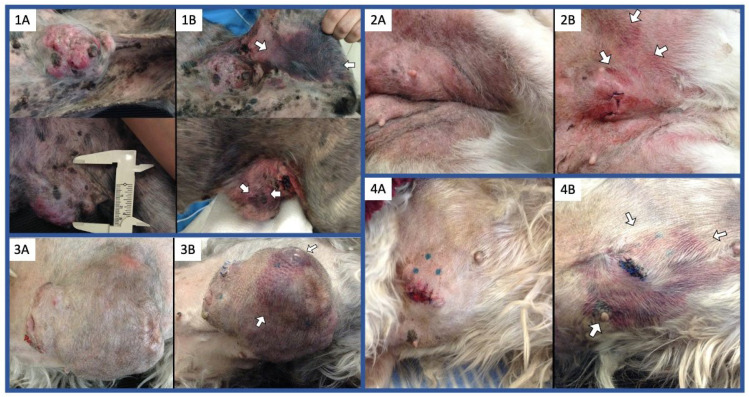
Local adverse event to NDV-MLS administration. Bruising around the injected tumor site and/or at the periphery of the injected tumor (white arrows) was noted in 4/6 dogs. Pictures pre-post (**A**) and post-virus (**B**) administration of the 4 affected patients (dogs 1–4) are shown.

**Table 1 viruses-16-00372-t001:** Female pet dogs enrolled in the study.

Dog	Breed	Castration Status	Age (Years)	Performance Status ^1^	Tumor Histologic Subtype/Histologic Grade	WHO Clinical Staging ^2^
1	Miniature Schnauzer	Spayed	10	2	Simple carcinoma/Intermediate	V (lung)
2	Mix breed	Spayed	11	2	Simple carcinoma/Intermediate	V (lung)
3	Miniature Schnauzer	Spayed	7	1	Tubular carcinoma/Intermediate	III
4	Miniature Schnauzer	Intact	12	1	Comedocarcinoma/High	V (lung)
5	Mix breed	Intact	11	1	Tubular carcinoma/Low	II
6	Labrador	Intact	10	1	Simple carcinoma/Intermediate	II
Control	Giant schnauzer	Intact	11	2	Simple carcinoma/Intermediate	At least III

^1^ The performance status defined as constitutional clinical signs grade ≤2 according to the Veterinary Cooperative Oncology Group—Common Terminology Criteria for Adverse Events (VCOG-CTCA v2) following investigational therapy in dogs and cats [40]. ^2^ WHO clinical staging for mammary cancer categorizes patients in stages I–V. I, tumor <3 cm; II, tumor 3–5 cm; III, tumor >5 cm; IV, any tumor size but presence of regional lymph node metastasis; V, any tumor size and lymph node status but evidence of distant metastasis [44].

**Table 2 viruses-16-00372-t002:** Coagulation test results of pet dogs who received the NDV-MLS. The values of the baseline (day 0) and at day 6 post-virus administration (day 6) are shown.

Dog	1		2		3		4		5		6	
Sample (Day)	0	6	0	6	0	6	0	6	0	6	0	6
Prothrombin time (PT) Reference interval: 7.0–9.4 s	7.4	6.6	6.8	6.5	6.9	7	7.4	8.1	7.5	7.9	8.1	7.9
Partial thromboplastin time (PTT) Reference interval: 8.5–13.8 s	20	17.6	13.7	13.9	12.3	12.2	12.8	17.4	16.2	15.9	14.6	15.2

**Table 3 viruses-16-00372-t003:** Serotype-specific antibodies on day 6 after NDV-MLS administration. Hemagglutination inhibition assay (HIA) in serum samples was performed in the presence of 4 hemagglutinating units (HAU) of NDV-MLS. The results are expressed as the total HAU in the serum samples.

Dog	HAU ^1^, Baseline (Day 0)	HAU ^1^ Day 6 Post-NDV-MLS
1	0	64
2	0	0
3	0	32
4	0	0
5	0	0
6	0	0

^1^ HAU: hemagglutinating.

**Table 4 viruses-16-00372-t004:** Percentage of necrosis, apoptosis, fibrosis, and cellular confluence on tumor samples at baseline (day 0) and day 6 post-viral administration. Dogs are identified from 1 to 6 and control dog (C).

D A Y 0	Necrosis							Apoptosis							Fibrosis							Cellular Confluence						
	C	1	2	3	4	5	6	C	1	2	3	4	5	6	C	1	2	3	4	5	6	C	1	2	3	4	5	6
	20	40	10	30	60	0	0	0	5	0	10	20	0	10	30	0	20	40	10	10	70	100	100	100	100	100	100	100
D A Y 6	Necrosis							Apoptosis							Fibrosis							Cellular Confluence						
	C	1	2	3	4	5	6	C	1	2	3	4	5	6	C	1	2	3	4	5	6	C	1	2	3	4	5	6
	90	0	NE	60	60	5	30	20	0	NE	20	30	5	10	70	0	NE	40	10	10	70	100	80	NE	70	90	80	100

**Table 5 viruses-16-00372-t005:** Percentage of immune cells in the different tumor compartments at baseline (day 0) and day 6 post-viral administration in the 6 female companion dogs (1–6). For lymphocytes, the percentage of CD3 and/or CD79 cells is included, except for the control dog (C).

D A Y 0	Tumor compartment		Stromal							Intratumoral							Peritumoral						
	Dog		C	1	2	3	4	5	6	C	1	2	3	4	5	6	C	1	2	3	4	5	6
	MNs	Lymphocytes			10	20				5	30										30	10	
		CD79a (+)			20	10					10										20	50	
		CD3 (+)			80	90					90										80	50	
		Both negative			0	0					0										0	0	
		Macrophages				10	10																
		Plasma cells																			30	10	
	PMNs	Neutrophils				10				20				10				10					
		Eosinophils																					
		Basophils																					
D A Y 6	Tumor compartment		Stromal							Intratumoral							Peritumoral						
	Dog		C	1	2	3	4	5	6	C	1	2	3	4	5	6	C	1	2	3	4	5	6
	MNs	Lymphocytes			60								40		5/10			5		10/30	40	10/10/20	30
		CD79a (+)			40								60		0/0			0		0/0	10	0/10/0	10
		CD3 (+)			60								40		50/80			100		50/10	90	20/90/80	90
		Both negative			0								0		50/20			0		50/90	0	80/0/20	0
		Macrophages				60	10	10					10							30		10	
		Plasma cells											30		10					40	40	10	30
	PMNs	Neutrophils				80							20	10		20							
		Eosinophils																					
		Basophils																					

MNs, mononuclear cells; PMNs, polymorphonuclear cells. In cases where multiple slides were evaluated, the minimum and maximum percentages that were observed are reported.

## Data Availability

Data are contained within the article and are also available upon request from the corresponding author.

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
