# Peer review of "The NDV-MLS as an Immunotherapeutic Strategy for Breast Cancer: Proof of Concept in Female Companion Dogs with Spontaneous Mammary Cancer"

_viruses, 2024, doi:10.3390/v16030372_

Round 1

Reviewer 1 Report

Comments and Suggestions for Authors

Certain details of the Materials and Methods are lacking or unclear:

o   What is the rationale for choosing Day 6 for post-tx assessment?

o   How were tumors measured and tumor volume calculated?

o   Was the “conventional surgery” inclusive of lymph node excision? If so, were they evaluated histologically? Were the tissue sections cut from completely excised masses?

o   Figure 5 provides some information about location of the Day 0 tumor biopsy sites. Was there uniformity regarding the site of the second tissue sample relative to location on the tumor, besides not in the same place as initial? Specifically, were samples primarily from the center or periphery of the tumor?

o   Line 245 – were peritumoral compartment samples from different tissue samples of the same mass? In other words, were they from the tumor/normal tissue interface whereas other compartment samples were taken more from the center of the mass?

o   Were pathologists blinded as to origin (pre- or post-tx; which animal) of the tissue being examined? Were pre- and post-tx samples from each animal compared side-by-side or evaluated and read out independently?

Results section

o   Line 375-Why were additional tissue samples obtained for dogs #3 and #5?

o   Measurement of tumor response via cRECIST v1.0 is mentioned in the M&M, but not reported. Why? If mentioned in Materials and Methods, should be addressed in Results and Discussion

Discussion

o   Lines 577-581 are important regarding limitations of the current study and logical next steps. Both #3 and #5 dogs had nonmetastatic tubular carcinoma and were the two with significant changes in tumor immune cell infiltrate.  It is important to address this and provide insight regarding the translatability of results from canine studies (with a variety of mammary tumor types with limited molecular characterization) to human clinical trials.

o   While beyond the scope of this proof-of-concept study, the experimental design did not facilitate comparison between intratumoral and systemic administration of oncolytic NDV-MLS.  This should be addressed in the discussion and in subsequent studies. Are both routes necessary and is there an effect on metastatic lesions?

Reviewer 2 Report

Comments and Suggestions for Authors

I really enjoyed the article, that seems original and with great clinical relevance. However, there are some aspects that I believe should be improved, namely:

in the introduction, it would be interesting to discuss reported adverse effects;

The 6 dogs included in this study were female companion dogs diagnosed with malignant mammary gland tumors, each having a longest diameter of ≥3cm. However, my concern is related to the specific subtype of the tumor that influenced significantly the population and the number of TILs. I think it would be interesting to classify the tumors by molecular subtype.

Finally, the NDV-MLS was injected at 3-5 points and avoided the site of the baseline biopsy. Could the biopsy procedure itself have contributed to inflammation and increased TILs? There is only one control animal, which appears to lack robustness.

Round 2

Reviewer 2 Report

Comments and Suggestions for Authors

Congratulations for the work.